# The Association between Participation in Organized Physical Activity and the Structure of Weekly Physical Activity in Polish Adolescents

**DOI:** 10.3390/ijerph18041408

**Published:** 2021-02-03

**Authors:** Dorota Groffik, Karel Frömel, Mateusz Ziemba, Josef Mitáš

**Affiliations:** 1Institute of Sport Sciences, The Jerzy Kukuczka Academy of Physical Education, Mikolowska 72A, 40-065 Katowice, Poland; d.groffik@awf.katowice.pl (D.G.); karel.fromel@upol.cz (K.F.); m.ziemba@awf.katowice.pl (M.Z.); 2Faculty of Physical Culture, Palacký University Olomouc, Tr. Miru 117, 77111 Olomouc, Czech Republic

**Keywords:** IPAQ-LF, physical activity structure, vigorous physical activity, gender differences, secondary school

## Abstract

The main aim of the study was to determine the associations of participation in organized physical activity (OPA), and the structure of weekly physical activity (PA) with meeting the PA recommendations among Polish boys and girls. The research was conducted between 2012 and 2019 in the Silesian region of Poland among 3499 secondary school students. To determine the structure of PA (school, transportation, home, recreation, vigorous moderate, and walking), the International Physical Activity Questionnaire-Long Form (IPAQ-LF) questionnaire was used. Adolescents participating in OPA showed significantly more PA (*p* < 0.001) than non-participating adolescents. The strongest associations were observed between participation in OPA and vigorous PA. The weekly recommendation of vigorous PA was met by 61% of the students with three or more lessons of OPA per week, 29% of students with one or two lessons of OPA per week, and 24% of students not participating in OPA. Therefore, boys and girls with no OPA are at greatest risk of health issues. Schools, sports clubs, and leisure institutions should increase the participation of adolescents in OPA, especially non-participants. Comprehensive school PA programs should especially include those forms of OPA that respect health weaknesses, individual talents for specific types of PA, and preferred types of PA among adolescents.

## 1. Introduction

Physical activity (PA) is the key factor influencing the health and quality of life of young people, including their school life [1]. Unfortunately, most studies indicate that children and adolescents in Europe [2], especially in Central Europe [3], do not meet the recommendations for PA [4,5,6,7]. Decreasing levels of PA are observed especially in adolescence [8,9,10,11,12]. The habit of regular PA instilled in youth will be effective in the later stages of their life and will reduce the likelihood of diseases, especially non-communicable diseases [13]. Therefore, Farooq et al. [14] emphasized that interventions aimed at preventing age-related declines in moderate to vigorous PA (MVPA) should be in place already at the beginning of adolescence.

The habit of regular PA in adolescents is, in addition to the family, dependent on high-quality school physical education (PE) and participation in organized PA (OPA). PE in Poland with three PE lessons per week (as opposed to the Czech Republic with two PE lessons per week) affords the bigger effects in secondary schools in terms of all-day vigorous PA (VPA) [15]. The VPA recommendation of at least 20 min three times a week is achieved by 39% of Polish (22% of Czech) boys and 35% of Polish (15% of Czech) girls.

A similar positive effect is caused by participation in OPA. In total, the VPA and MVPA recommendations are met by 29.6% of Czech and Polish adolescents actively participating in OPA, but only by 14.9% of non-participating adolescents [16]. The lowest degree of achievement of PA recommendations was reported by Lagestad et al. [17] in 12–13-year-old boys who participated in <3 h per week (or not at all) of organized sport. Adolescents’ participation in OPA is associated with a higher level of MVPA and VPA, as well as meeting the MVPA recommendations [18], among girls [19] and not just boys. Similar positive conclusions were also formulated by Jago et al. [20] in children. Boys’ and girls’ active participation in OPA in school and community settings is associated with greater PA and reduced sedentary time. Adolescents’ participation in OPA is significantly associated with greater health benefits [21,22], as well as school performance [23], compared to engagement in unstructured activities. On the contrary, these activities are associated with greater health risks and risk behavior among adolescents [24].

Participation in OPA also improves adolescents’ cardiorespiratory fitness [25] and body composition [26]. However, the associations between children’s and adolescents’ participation in OPA and body mass index (BMI) are not entirely clear [27]. There are also considerations as to whether the inclusion of inactive children in OPA will bring the same long-term effects as in the case of participating children [28].

Therefore, we believe it is important to analyze the associations between participation in OPA and the structure of weekly PA in order to ensure more effective participation in OPA. The following research questions were formulated: What are the differences in the weekly PA structure between boys and girls, including the eight-year monitoring? What is the degree of PA achievement among boys and girls with different levels of participation in OPA? The main aim of this study was to determine the associations of participation in OPA, and the structure of weekly PA with meeting the PA recommendations among Polish boys and girls.

## 2. Materials and Methods

### 2.1. Participants and Setting

In total, 3499 secondary school students (1576 boys and 1923 girls) from the Silesian–Katowice and Wroclaw regions in Poland took part in the research (Table 1). This study was carried out between 2012 and 2019 in 89 Polish schools using the “International Database for Research and Educational Support” (Indares) web application (www.indares.com). Members of the school administration, parents, and participants gave their written informed consent to participate in this research, which was carried out as part of the school program.

The study was conducted in accordance with the Declaration of Helsinki, and was approved by the Ethical Committee of The Jerzy Kukuczka Academy of Physical Education in Katowice under Reg. No. 2/2008. Participants self-reported a healthy status and no restrictions to completing the research protocol.

### 2.2. Procedures

This research was conducted in public schools, most of which cooperate with The Jerzy Kukuczka Academy of Physical Education in Katowice as part of mid-year internships involving students of PE. After obtaining permission for the research study, the purpose and scope of the research was presented in each class. The participants then registered via the Indares.com web application, where they completed the International Physical Activity Questionnaire-Long Form (IPAQ-LF). The researchers provided instructions on how to correctly complete the questionnaire and explained the purpose of the questions if necessary.

### 2.3. Measurements

The structure of the participants’ weekly PA was examined using the IPAQ-LF [29] for adolescents [30,31]. The Polish version was subject to the required translation procedure according to the European Organization for Research and Treatment of Cancer Quality of Life Group [32] and empirically verified in international comparative studies [33,34]. Our experience and empiric results indicated an overestimation of the time spent in PA and an underestimation of the time spent sitting [35]. Our procedure for data adjustments according to the IPAQ-LF manual was as follows: Compared with the IPAQ-LF manual, the MET-min of VPA was multiplied by six instead of the recommended multiplication by eight. The permissible average daily sum of PA and transportation was set at 600 min. The maximum MET-min per week was set at 16,000, and the maximum average daily sum of PA, transportation, sitting, and passive commuting was set at 960 min. A total of 133 respondents were excluded due to incomplete data.

The recommendations of weekly PA were modified according to Healthy People 2020 [36] and the Physical Activity Guidelines for Americans [37]. The minimum recommendation was selected because it was based only on a single given type of PA in the IPAQ-LF questionnaire. The weekly PA recommendations are as follows: At least five or more days of at least 30 min of moderate PA (MPA) per week (5 × 30 min MPA), five days of 30 min of walking per week (5 × 30 min walking), five days of 60 min of MVPA per week (5 × 60 min MVPA), and three days of 20 min of VPA per week (3 × 20 min VPA). This also includes a combination of 3 × 20 VPA and 5 × 60 MVPA (5 × 60 min MVPA + 3 × 20 min VPA). The eight-year research period was divided into four two-year stages (2012–2013, 2014–2015, 2016–2017, and 2018–2019). Irrespective of gender, the participants were divided according to their participation in OPA into three groups: Participation in three or more OPA lessons per week (≥3 lessons), participation in one or two OPA lessons per week (1–2 lessons), and non-participation in OPA (0 lessons). OPA lessons included school PE lessons and all forms of organized and structured PA with a leader.

### 2.4. Data Analysis

Statistica version 13 (StatSoft, Prague, Czech Republic) and SPSS version 25 (IBM Corp., Armonk, NY, USA) were used for the purposes of statistical analysis. Descriptive characteristics and crosstabulation tables were used to assess the differences in compliance with the PA recommendations. Gender-based OPA was assessed using the chi-squared and Mann–Whitney *U* tests. The Kruskal–Wallis ANOVA was applied to the IPAQ-LF results. Practical significance was evaluated using the *η^2^* and *r* effect size coefficients and interpreted as follows: Small effect, 0.01 ≤ *η^2^* < 0.06 (0.1 ≤ *r* < 0.2); medium effect, 0.06 ≤ *η^2^* < 0.14 (0.2 ≤ *r* < 0.6); large effect, *η^2^* ≥ 0.14 (*r* ≥ 0.6). The level of significance was set at *p* < 0.05.

## 3. Results

### 3.1. Differences in the Structure of Weekly PA Among Boys and Girls

The boys were more physically active than the girls in school (*p* = 0.016), within recreation (*p* = 0.001), vigorous PA (*p* < 0.001), and total PA (*p* < 0.001). Meanwhile, the girls were more physically active than the boys in terms of walking (*p* < 0.001) (Table 2). However, considering the two-year stages (2012–2013, *p* = 0.002; 2014–2015, *p* = 0.020; 2016–2017, *p* < 0.001; 2018–2019, *p* = 0.099), VPA was the only type where the boys reported a significantly greater amount. In the remaining types of weekly PA, the differences by gender were not significant.

Between the first (2012–2013) and last stage (2018–2019), there was a statistically significant decrease in school PA among boys (*p* = 0.005) and girls (*p* = 0.009). In girls, similar differences were observed in transportation PA (p = 0.021), recreation PA (*p* < 0.001), and walking (*p* = 0.002).

### 3.2. Differences in the Structure of Weekly PA by Participation of Boys and Girls in OPA

In total, for boys and girls irrespective of gender, those who participated in OPA in three or more lessons per week showed statistically significantly more PA in all types of PA than those who did not participate (Table 3). As expected, the biggest differences between adolescents participating or not in OPA was in school, recreation, and vigorous PA. Even those who participated in one or two lessons of OPA reported more school (*p* = 0.002), home (*p* = 0.001), recreation (*p* < 0.001), vigorous (*p* < 0.001), moderate (*p* < 0.001), and total PA (*p* < 0.001) compared to those who did not participate.

Boys participating in OPA in three or more lessons per week reported more school (*p* < 0.001), recreation (*p* < 0.001), vigorous (*p* < 0.001), moderate (*p* < 0.001), walking (*p* = 0.006), and total PA (*p* < 0.001) compared to non-participating boys (Table 4). Girls participating in OPA in three or more lessons per week reported more school (*p* < 0.001), recreation (*p* < 0.001), vigorous (*p* < 0.001), moderate (*p* < 0.001), and total PA (*p* < 0.001) than non-participating girls. Even those boys who participated in one or two lessons of OPA per week reported more recreation (*p* < 0.001), vigorous (*p* < 0.001), moderate (*p* < 0.001), and total PA (*p* < 0.001) than non-participating boys. Similarly, girls participating in one or two lessons of OPA per week reported more school (*p* = 0.022), recreation (*p* < 0.001), vigorous (*p* < 0.001), moderate (*p* = 0.009), and total PA (*p* = 0.004) than non-participating girls.

Remarkable differences were observed between boys and girls with identical participation in OPA. Only in vigorous PA (*p* < 0.001) and total PA (*p* = 0.007) were boys with participation in OPA in three or more lessons physically more active than girls with the same participation in OPA. Boys with participation in OPA in one or two lessons were physically more active in moderate PA (*p* = 0.009) than girls with the same participation in OPA.

### 3.3. Differences in the Structure of PA by Participation of Boys and Girls in OPA by Two-Year Stages

In all of the two-year stages, boys with participation in three or more lessons of OPA per week had statistically significantly more recreation PA (*p* < 0.001), vigorous PA (*p* < 0.001), and overall weekly PA (*p* = 0.010) than boys not participating in OPA. Girls with participation in three or more lessons of OPA per week had statistically significantly more recreation PA (*p* = 0.010), vigorous PA (*p* < 0.001), and overall weekly PA (except 2012–2013) (*p* = 0.010) than girls not participating in OPA.

In the course of the research, no decrease was observed in the participation in OPA among boys (38.6% in 2012–2013 and 41.1% in 2018–2019) or in girls (29.0% in 2012–2013 and 32.6% in 2018–2019).

### 3.4. Association between Participation of Boys and Girls in OPA and Achievement of PA Recommendations

Adolescents with participation in three or more lessons of OPA per week showed a statistically significantly higher achievement of the recommendations for VPA (3 × 20 min) (*p* < 0.001), MPA (5 × 30 min) (*p* < 0.001), walking (5 × 30 min) (*p* < 0.001), MVPA (5 × 60 min) (*p* < 0.001), and combined MVPA and VPA (5 × 60 min + 3 × 20 min) (*p* < 0.001) than adolescents not participating in OPA (Figure 1). Adolescents with participation in one or two lessons of OPA per week showed a statistically significantly higher achievement of the recommendations only for VPA (3 × 20 min) (*p* = 0.011) than adolescents not participating in OPA.

Boys participating in three or more lessons of OPA per week showed a statistically significantly higher achievement of the recommendations for VPA (χ^2^ = 141.63, *p* < 0.001; *r* = 0.299), walking (χ^2^ = 6.31, *p* = 0.012; *r* = 0.062), MVPA (χ^2^ = 22.60, *p* < 0.001; *r* = 0.118), and combined MVPA and VPA (χ^2^ = 81.58, *p* < 0.001; *r* = 0.227) than boys not participating in OPA (Figure 2). Girls participating in three or lessons of OPA per week showed a statistically significantly higher achievement of the recommendations for VPA (χ^2^ = 114.38, *p* < 0.001; *r* = 0.246), MPA (χ^2^ = 10.99, *p* < 0.001, *r* = 0.072), MVPA (χ^2^ = 9.56, *p* = 0.002; *r* = 0.068), and combined MVPA and VPA (χ^2^ = 71.01, *p* < 0.001; *r* = 0.192) than girls not participating in OPA.

Boys participating in three or more lessons of OPA per week were more likely (OR = 4.16, 95% CI = 3.02–5.75, *p* < 0.001) to meet the most stringent PA recommendation (currently 5 × 60 min MVPA + 3 × 20 min VPA) compared to boys not participating in OPA. Similarly, girls participating in three or more lessons of OPA per week were more likely to meet this PA recommendation than girls not participating in OPA (OR = 3.50, 95% CI = 2.59–4.73, *p* < 0.001). The control variables of age, BMI, and two-year research stages did not affect the significance of the likelihood of meeting the PA recommendations by boys or girls participating in OPA.

## 4. Discussion

The most significant finding of the study was that adolescents who participated in three or more lessons of OPA were more physically active in all monitored types of PA, as well as total weekly PA. In addition to transportation PA and walking, this finding also applies to adolescents with participation in one or two lessons of OPA. The fact that participation in OPA increases overall PA has been confirmed by many studies in children [38] and in adolescents [39]. This mainly applies to the association between participation in OPA and VPA [40]. Less knowledge is available regarding the association between participation in OPA and active transportation or walking. Santos et al. [41] did not find significant associations between participation in OPA and active traveling in Portuguese adolescents. Surprisingly, the association between OPA and walking was confirmed by the present study only in boys. The differences in walking between boys and girls participating in OPA were not statistically significant, while in weekly PA (regardless of participation in OPA) girls show more walking than boys.

In similar conditions in the Czech Republic, no significant differences were observed in walking between adolescent boys and girls [42]. However, during the specific times like the COVID–19 pandemic, walking should be significantly represented in OPA.

A very positive aspect is that participation in OPA increased the weekly PA in most types of PA in both the boys and girls, which we consider to be a completely new finding. The contribution to the increase in PA in different types of PA may also be significant for PA in adulthood. Telama et al. [43] have already pointed out that participation in youth OPA predicts PA in adulthood.

A less positive finding is the low school PA in boys and girls not participating in PE lessons. Moreover, recreation PA in terms of MET-min was lower than school PA in all groups by participation in OPA. This is consistent with the benefits of three PE lessons per week as defined by the curriculum, but also with the lower participation in OPA among Polish adolescents [44]. The effects of the higher number of PE lessons on the increase in weekly vigorous PA seem to be crucial in the justification of PE in schools [15]. This is also supported by the evidence that PE lessons contribute to increased MVPA in children [45] and adolescents [15,18,46].

The achievement of the weekly PA recommendations corresponds with the associations between participation in OPA and the types of PA. In particular, the recommendation for VPA, as well as the combination of at least 5 × 60 min MVPA and 3 × 20 min VPA, were significantly associated with participation in OPA in the boys and girls. This finding is consistent with the previous results in the Central European Region, as well as other countries [40]. Participation in OPA of 7–12-year-old children three or more times a week has been shown to increase the odds of meeting PA recommendations [47]. Children that participated in organized PA also have higher odds of meeting the combination of PA and sleep recommendations [48].

The importance of participation in OPA lies not only in supporting the habit of regular PA, but also in maintaining and increasing cardiovascular endurance [49]. The habit of regular OPA should also be maintained by emphasizing the awareness of emotions, feelings, and well-being [50]. OPA, especially school PA, as well as other structured forms of PA, are greatly responsible for acquiring motor skills and physical literacy in adolescents.

The fact that the recommendation for VPA of 3 × 20 min was achieved by 66% of the boys (55% of the girls) and the combination of 5 × 60 min MVPA and 3 × 20 min VPA was achieved by 46% of the boys (39% of the girls) allows to consider that the current PA recommendations are adequate regarding the overall decrease in achievement. Therefore, it is essential to increase the involvement of Polish youths in OPA after the pandemic. According to the previous findings, the participation of Polish youth in OPA does not match the level of participation of young people in the Czech Republic, Sweden, or other European countries [8]. Therefore, it is necessary to verify the objective possibilities of comparing the participation of adolescents in OPA in European countries, as the methodology of Global Matrix 3.0 initiative was not wholly accepted [51]. Special emphasis should be placed on greater involvement of girls in OPA because, according to this study, they achieve a statistically significantly lower amount of vigorous and total PA than boys. This may be partly related to the lower involvement of girls in three or more lessons of OPA (30.9%) compared to boys (41.1%).

The benefit of adolescents’ participation in OPA is significant not only from a health aspect, but also from educational, social, and psychological perspectives. For example, participation of children and adolescents in OPA, especially in team sports, brings psychological and social benefits [52,53]. Participation in team sports is strongly associated with improving mental health [54]. Indeed, Polish adolescents significantly prefer team sports and participation in team sports in the context of OPA [16].

The involvement of hitherto non-participating adolescents in OPA after school requires respect for the diverse interests and activities of adolescents and efforts to link them with a suitable type of OPA [55]. It should also be respected that 63% of adolescents in Polish schools participate in PE mainly for “fun–pleasure–entertainment” [56]. The fact that four in every five adolescents do not experience the enjoyment or social, physical, and mental health benefits of regular PA is caused especially by the inability to participate in OPA, which is largely dependent on socioeconomic and political decisions [10].

There is a need to improve cooperation between schools, sports clubs, and leisure institutions that focus on PA, as well as to provide the conditions for equal access of all adolescents to regular participation in PA. At the school level, effective cooperation should be promoted by means of high-quality comprehensive physical activity programs. However, in addition to institutional support for OPA participation, a crucial aspect is the socioeconomic support of families and friends [57] and society-wide priority in promoting a healthy lifestyle for all population groups.

The strength of this study is its investigation of the associations between OPA and the structure and intensity of weekly PA in adolescents over a period of eight years. In the schools involved, the research was performed by the same research team using the Indares web-based application.

The limitations of the research study include the deliberate sampling of adolescents and the analysis of the weekly compositions and structure of weekly PA based only on the IPAQ-LF questionnaire. Another limitation is that for the purposes of simplification, OPA included all structured PA with a leader, including PE lessons, extracurricular OPA in schools, and organized PA lessons in free time.

## 5. Conclusions

Participation in OPA increases the school, transportation, home, recreation, walk, moderate, vigorous, and total weekly physical activity levels of adolescents. The strongest associations were observed between participation in OPA and vigorous PA in both boys and girls. The group at greatest risk comprises boys and girls with no OPA, no PE lessons, and no OPA lessons. Comprehensive school PA programs in cooperation with leisure institutions should also include organized forms of PA that support the increase of PA in all adolescent boys and girls. However, increased attention should be paid to adolescents who do not participate in any OPA and tend to have health or recreation-oriented PA. It is also necessary to support preferred types of PA in the OPA.

## Figures and Tables

**Figure 1 ijerph-18-01408-f001:**
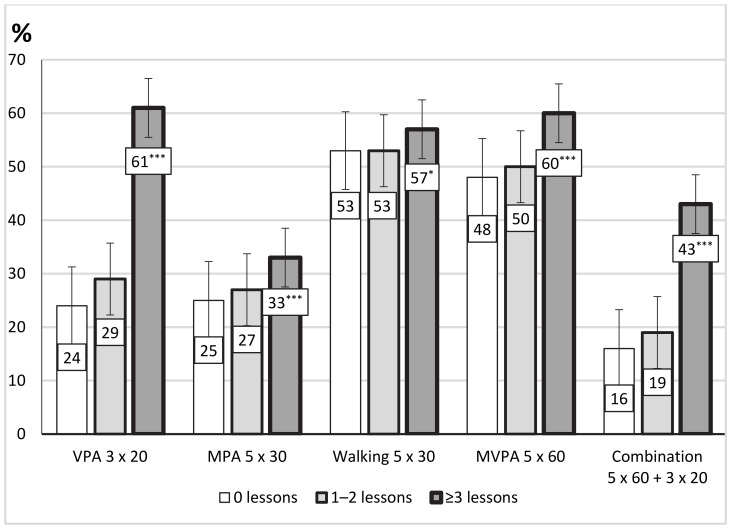
Achievement of the recommendations for weekly physical activity(PA) (3 × 20 min of vigorous PA (VPA), 5 × 30 min of moderate PA (MPA), 5 × 30 min of walking, 5 × 60 min of moderate to vigorous PA (MVPA), and 5 × 60 min of MVPA combined with 3 × 20 min of VPA) by adolescents’ participation in organized physical activity. Significant difference between groups (0 lessons and ≥3 lessons of OPA): * *p* < 0.05; *** *p* < 0.001.

**Figure 2 ijerph-18-01408-f002:**
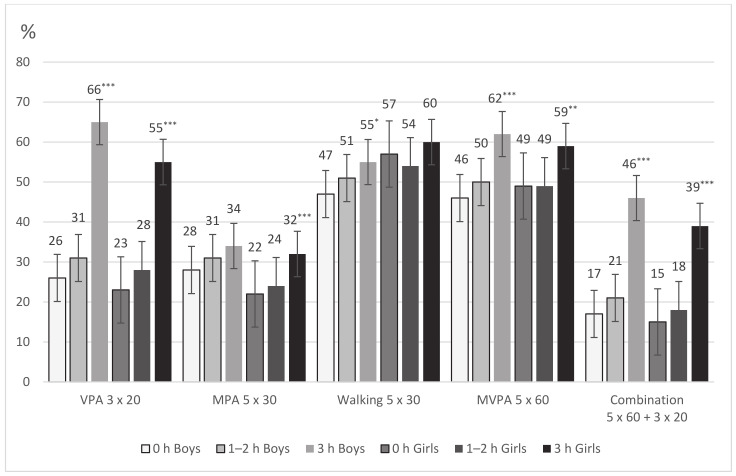
Achievement of the recommendations for weekly physical activity (3 × 20 min of vigorous PA (VPA), 5 × 30 min of moderate PA (MPA), 5 × 30 min of walking, 5 × 60 min of moderate to vigorous PA (MVPA), and 5 × 60 min of MVPA combined with 3 × 20 min of VPA) by boys’ and girls’ participation in organized physical activity. Significant difference between groups (0 lessons and ≥3 lessons of OPA): * *p* < 0.05; ** *p* < 0.01; *** *p* < 0.001.

**Table 1 ijerph-18-01408-t001:** Sample characteristics.

Characteristics	Stages	*n*	Age (Years)		Weight (kg)		Height (cm)		BMI (kg·m^−2^)	
			M	SD	M	SD	M	SD	M	SD
Boys	2012–2013	420	16.36	0.78	67.05	11.17	177.77	6.94	21.18	3.12
	2014–2015	371	16.24	0.80	67.39	12.45	177.56	7.02	21.32	3.40
	2016–2017	510	16.62	0.87	69.18	12.56	178.27	7.58	21.70	3.32
	2018–2019	275	16.15	0.75	68.01	11.84	176.45	7.60	21.77	3.12
Girls	2012–2013	483	16.28	0.78	57.05	9.32	165.92	5.82	20.69	2.99
	2014–2015	574	16.44	0.65	56.81	8.54	166.36	6.12	20.51	2.75
	2016–2017	504	16.49	0.81	57.44	9.05	165.84	6.24	20.86	2.94
	2018–2019	362	16.31	0.77	56.81	9.10	164.66	6.15	20.94	3.07

Notes: BMI, body mass index; M, mean; SD, standard deviation.

**Table 2 ijerph-18-01408-t002:** Types of weekly physical activity (MET-min·week^−1^) among Polish boys and girls using the International Physical Activity Questionnaire-Long Form (IPAQ-LF).

PA (MET-Min·Week^−1^)	Gender				*U*	*p*	*η^2^*
	Boys (*n* = 1576)		Girls (*n* = 1923)				
	Mdn	IQR	Mdn	IQR			
School	1420	3464	1077	2990	2.42	0.016	0.004
Transportation	825	1863	693	1749	1.12	0.264	0.001
Home	385	1000	420	910	0.91	0.364	0.001
Recreation	773	2233	672	1580	3.22	0.001	0.007
Vigorous	1080	2790	540	1860	7.69	0.000	0.038 *
Moderate	1618	2940	1238	2188	4.85	0.000	0.015 *
Walking	1238	2731	1485	2789	3.75	0.000	0.009
Total	5331	6589	4404	6109	4.43	0.000	0.012 *

Notes: PA, physical activity; Mdn, median; IQR, interquartile range; *U*, Mann–Whitney test; *p*, significance level; *η^2^*, effect size coefficient; *, small effect size.

**Table 3 ijerph-18-01408-t003:** Weekly physical activity (MET-min·week^−1^) among adolescents according to participation in organized physical activity, assessed using the IPAQ-LF.

PA (MET-Min·Week^−1^)	Number of Lessons of Organized Physical Activity Per Week						*H*	*p*	*η^2^*
	0 Lessons (*n* = 810)		1–2 Lessons (*n* = 1448)		≥3 Lessons (*n* = 1241)				
	Mdn	IQR	Mdn	IQR	Mdn	IQR			
School	803	2168	1129	3040	1755	3820	67.38 ^a,b^	<0.001	0.019 *
Transportation	693	1580	693	1848	924	1866	14.38 ^a^	<0.001	0.004
Home	340	780	450	960	420	985	15.46 ^a,b^	<0.001	0.004
Recreation	300	1098	628	1484	1432	2702	290.37 ^a,b^	<0.001	0.082 **
Vigorous	60	900	600	1800	1740	3180	437.59 ^a,b^	<0.001	0.125 **
Moderate	940	2145	1333	2355	1740	2935	70.79 ^a,b^	<0.001	0.020 *
Walking	1271	2508	1,337	2855	1584	2904	14.39 ^a^	<0.001	0.004
Total	3356	4780	4459	5987	6367	6772	202.85 ^a,b^	<0.001	0.057 *

Notes: PA, physical activity; Mdn, median; IQR, interquartile range; *H*, Kruskal–Wallis test; *p*, significance level; *η^2^,* effect size coefficient; *, small effect size; **, medium effect size. ^a^ Significant difference between groups (0 lessons and ≥3 lessons of organized physical activity). ^b^ Significant difference between groups (0 lessons and 1–2 lessons of organized physical activity).

**Table 4 ijerph-18-01408-t004:** Weekly physical activity (MET-min·week^−1^) among boys and girls according to participation in organized physical activity (IPAQ-LF).

Physical Activity (MET-Min·Week^−1^)	Number of Lessons of Organized Physical Activity Per Week						*H*	*p*	*η^2^*
	Boys 0 Lessons	Boys 1–2 Lessons	Boys ≥3 Lessons	Girls 0 Lessons	Girls 1–2 Lessons	Girls ≥3 Lessons			
	Mdn (IQR)	Mdn (IQR)	Mdn (IQR)	Mdn (IQR)	Mdn (IQR)	Mdn (IQR)			
School	840 (2606)	1220 (3097)	2010 (3957)	756 (2071)	1050 (2943)	1458 (3620)	75.85 ^a,b,d^	<0.001	0.020 *
Transportation	743 (1550)	773 (2094)	924 (1866)	693 (1569)	683 (1617)	924 (1934)	16.99	0.004	0.003
Home	320 (830)	455 (1105)	375 (970)	360 (740)	450 (885)	450 (1010)	19.28 ^b^	0.002	0.004
Recreation	264 (858)	693 (1631)	1709 (2928)	339 (1139)	592 (1373)	1194 (2354)	299.57 ^a,b,c,d^	<0.001	0.084 **
Vigorous	120 (1080)	720 (2160)	2280 (3510)	0 (810)	525 (1680)	1290 (2550)	479.51 ^a,b,c,d^	<0.001	0.136 **
Moderate	1258 (2563)	1603 (3090)	1870 (2990)	795 (1865)	1260 (1920)	1650 (2945)	90.17 ^a,b,c,d^	<0.001	0.024 *
Walking	1023 (2178)	1188 (2805)	1436 (3069)	1403 (2558)	1394 (2937)	1733 (2789)	32.68 ^a^	<0.001	0.008
Total	337(4830)	4783 (6338)	7001 (6776)	3335 (4788)	4215 (5690)	5666 (6773)	217.14 ^a,b,c,d^	<0.001	0.061

Notes: Mdn, median; IQR, interquartile range; *H*, Kruskal–Wallis test; *η^2^*, effect size coefficient; *p*, significance level; *, small effect size; **, medium effect size. Significant difference between groups: ^a^ Boys 0 lessons and boys ≥3 lessons, ^b^ girls 0 lessons and girls ≥3 lessons, ^c^ boys 0 lessons and boys 1–2 lessons, and ^d^ girls 0 lessons and girls 1–2 lessons.

## Data Availability

The data presented in this study are available on request from the corresponding author.

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
