# Peer review of "The Association between Participation in Organized Physical Activity and the Structure of Weekly Physical Activity in Polish Adolescents"

_ijerph, 2021, doi:10.3390/ijerph18041408_

Round 1
Reviewer 1 Report
Review "The Association between Participation in Organized Physical Activity and the Structure of Weekly Physical Activity in Polish Adolescents" .
The article is another example of showing the increase in physical activity of young people, where there are additional classes at school and in free time, organized by sports clubs, associations, etc. - and as the authors write ... Physical activity is the key factor influencing the health and quality of life of 29 young people, including their school life. Abstract - OK Admission The introduction is written briefly and addresses important issues for the article. The authors have analyzed very well various publications in the field of physical activity, divided into organized, compulsory and free time. In poems No. 39-42 the authors write ... "PE in Poland with three PE lessons per week (as opposed to the Czech Republic with two PE lessons per week) affords the greatest effects in secondary schools in terms of all-day vigorous PA (VPA) "... Does it mean that these are greater effects in Poland as opposed to the Czech Republic - or to other countries? - I am asking because the word "greatest" was used? - suggests changing the word to "bigger" Materials and Methods Participants and Setting - approx Measurements In lines 103-104 the authors write that ... A total of 133 respondents were excluded due to incomplete data ... - is the number in table 1 consistent with this group or without it? Results Differences in the Structure of Weekly PA Among Boys and Girls - OK Differences in the Structure of Weekly PA by Participation of Boys and Girls in OPA - approx Differences in the Structure of PA by Participation of Boys and Girls in OPA by Two-Year Stages - approx Association between Participation of Boys and Girls in OPA and Achievement of PA 194 Recommendations - approx Discussion In lines 240-241 the authors write that ... "One of the reasons is that girls showed less walking than boys in all of the stages ..." - while in lines 132-133 the authors write that ... "Meanwhile , the girls were more physically active than the boys in terms of walking ... "- there is some incompatibility here ... Conclusions He suggests conducting an analysis with motion sensors, e.g. with accelerometers to check (analyze) overestimations in the physical activity reported by young people - you can see the WGT3X study, e.g. Herbert, J. Matłosz, P. Lenik, J. Szybisty, A. Baran, J. Przednowek , K. Wyszyńska J. Objectively Assessed Physical Activity of Preschool-Aged Children from Urban Areas, IJERPH, 2020, Vol. 17,4, no. 1375, DOI: 10.3390 / ijerph17041375 References No. 4 without underlining No. 8 without underlining No. 17 without underlining No. 28 without underlining Entry # 29 starts with ... Craig, C.L .; Marshall, A.L .; Sjöström, M .; Bauman, A.E .; Booth, M.L .; Ainsworth, B.E .; Pratt, M .; Ekelund, U .; Yngve, A .; Sallis, 396JF .; et al. International physical activity questionnaire: 12-country reliability and validity. Med. Sci. Sports Exerc. 2003. 35. 1381– 397 1395. doi: 10.1249 / 01.MSS.0000078924.61453.FB - and should from CORA L ...
Author Response
Comments and Suggestions for Authors
Review "The Association between Participation in Organized Physical Activity and the Structure of Weekly Physical Activity in Polish Adolescents" .
The article is another example of showing the increase in physical activity of young people, where there are additional classes at school and in free time, organized by sports clubs, associations, etc. - and as the authors write ... Physical activity is the key factor influencing the health and quality of life of 29 young people, including their school life. Abstract - OK Admission The introduction is written briefly and addresses important issues for the article. The authors have analyzed very well various publications in the field of physical activity, divided into organized, compulsory and free time. In poems No. 39-42 the authors write ... "PE in Poland with three PE lessons per week (as opposed to the Czech Republic with two PE lessons per week) affords the greatest effects in secondary schools in terms of all-day vigorous PA (VPA) "... Does it mean that these are greater effects in Poland as opposed to the Czech Republic - or to other countries? - I am asking because the word "greatest" was used? - suggests changing the word to "bigger"
Response:
Dear reviewer, we would like to thank you for your feedback and valuable recommendations. We revised the paper based on your recommendations and important points. In previous research, we found that three PE lessons in Poland are manifested mainly in more weekly VPA, but not in the total weekly PA, and we wanted to mention it here, which is a serious finding. You are right and we've used bigger instead of greatest.
Materials and Methods Participants and Setting - approx Measurements In lines 103-104 the authors write that ... A total of 133 respondents were excluded due to incomplete data ... - is the number in table 1 consistent with this group or without it?
Response:
The table 1 contains only final sample of the participants. We moved the information about the exclusion of students into measurements part.
Results Differences in the Structure of Weekly PA Among Boys and Girls - OK Differences in the Structure of Weekly PA by Participation of Boys and Girls in OPA - approx Differences in the Structure of PA by Participation of Boys and Girls in OPA by Two-Year Stages - approx Association between Participation of Boys and Girls in OPA and Achievement of PA 194 Recommendations - approx Discussion In lines 240-241 the authors write that ... "One of the reasons is that girls showed less walking than boys in all of the stages ..." - while in lines 132-133 the authors write that ... "Meanwhile , the girls were more physically active than the boys in terms of walking ... "- there is some incompatibility here ...
Response:
Thank you for the comment. We've edited the text to make it clearer: „The differences in walking between boys and girls participating in OPA were not statistically significant, while in weekly PA (regardless of participation in OPA) girls show more walking than boys.“
Conclusions He suggests conducting an analysis with motion sensors, e.g. with accelerometers to check (analyze) overestimations in the physical activity reported by young people - you can see the WGT3X study, e.g. Herbert, J. Matłosz, P. Lenik, J. Szybisty, A. Baran, J. Przednowek , K. Wyszyńska J. Objectively Assessed Physical Activity of Preschool-Aged Children from Urban Areas, IJERPH, 2020, Vol. 17,4, no. 1375, DOI: 10.3390 / ijerph17041375
Response:
Thank you for the comment. We know about this publication, but due to the age differences of the participants, we did not use it.
References No. 4 without underlining No. 8 without underlining No. 17 without underlining No. 28 without underlining Entry # 29 starts with ... Craig, C.L .; Marshall, A.L .; Sjöström, M .; Bauman, A.E .; Booth, M.L .; Ainsworth, B.E .; Pratt, M .; Ekelund, U .; Yngve, A .; Sallis, 396JF .; et al. International physical activity questionnaire: 12-country reliability and validity. Med. Sci. Sports Exerc. 2003. 35. 1381– 397 1395. doi: 10.1249 / 01.MSS.0000078924.61453.FB - and should from CORA L .??..
Response:
Thank you for the comment. We revised these typos and also removed redundant information.
Reviewer 2 Report
Dear authors,
After reviewing your manuscript, I think that some points in the manuscript should be revised. Next, I will list my comments:
Abstract
The summary shows “Therefore, boys and girls with no OPA are at geatest risk of health issues”, but nothing above about health benefits appears in the previous results. I would understand, that it was put as an introduction, as if physical activity reduces health problems.
Another statement such as “Schools, sports clubs and leisure institution should focus on students not participating in OPA”. I think they should focus on all adolescents, although with special attention to those who do not participate in OPA.
Keywords
I think the keyword "Recommendation" is a very general word and it would be difficult to classify this manuscript with that word. In addition, the keyword IPAQ-LF would be more valid and not IPAQ.
Material and Methods.
2.3. Measurements
Line 103: “A total of 133 respondents were excluded due to incomplete data”
Wouldn't it be better in results?
If in Poland, the OPA is 3 hours, how can the other groups be obtained? Are those 3 hours a week optional?
Results
It seems logical that adolescents with participation in three or more lessons of OPA per week, met the VPA recommendations. They already had activities scheduled and with them they could fulfill those recommendations.
Discussion
The discussion seems more like an introduction to me, since there is no comparison between its results and those obtained by other researchers.
Although with a different objective, in a previous publication*, they already found differences between two educational systems, which differed in physical education lessons.
Conclussions: Participation in PELs was associated with a higher rate of meeting SPA recommendations in both countries. Compared with the Czech Republic, more PELs in the Polish education system was associated with increased daily vigorous PA and a greater portion of SPA in daily PA. Differences in overall daily and weekly moderate-to-vigorous PA between Polish and Czech adolescents were non-significant.
What does this new study contribute?
Kind regards.
*Groffik D, Mitáš J, Jakubec L, Svozil Z, Frömel K. Adolescents' Physical Activity in Education Systems Varying in the Number of Weekly Physical Education Lessons. Res Q Exerc Sport. 2020;91(4):551-561. doi:10.1080/02701367.2019.1688754
Author Response
Comments and Suggestions for Authors
Dear authors,
After reviewing your manuscript, I think that some points in the manuscript should be revised. Next, I will list my comments:
Abstract
The summary shows “Therefore, boys and girls with no OPA are at geatest risk of health issues”, but nothing above about health benefits appears in the previous results. I would understand, that it was put as an introduction, as if physical activity reduces health problems.
Response:
Dear reviewer, we would like to thank you for your feedback and valuable recommendations. We revised the paper based on your recommendations and important points. We also edited the aim: “The main aim of this study was to determine the associations of participation in OPA, and the structure of weekly PA with meeting the PA recommendations among Polish boys and girls.”
Another statement such as “Schools, sports clubs and leisure institutions should focus on students not participating in OPA”. I think they should focus on all adolescents, although with special attention to those who do not participate in OPA.
Response:
We agree. Knowing the state and trends in PA of adolescents, it is necessary to pay attention to all adolescents. We changed the wording”: “Schools, sports clubs and leisure institutions should increase the participation of adolescents in OPA, especially non-participants”
Keywords
I think the keyword "Recommendation" is a very general word and it would be difficult to classify this manuscript with that word. In addition, the keyword IPAQ-LF would be more valid and not IPAQ.
Response:
We adapted the keywords: IPAQ-LF; physical activity structure; vigorous physical activity; gender differences; secondary school
Material and Methods.
2.3. Measurements
Line 103: “A total of 133 respondents were excluded due to incomplete data”
Wouldn't it be better in results?
Response:
The information about exclusion of participants is important for the evaluation of the study results. However, it is part of introduction in the methods part of the paper structure.
If in Poland, the OPA is 3 hours, how can the other groups be obtained? Are those 3 hours a week optional?
Response:
In the Polish curriculum, three PE lessons per week are compulsory for secondary schools. Exemption for health reasons or excuse from participation is possible. It is difficult to distinguish between participation in PE and other forms of OPA, so we consider it most objective to divide groups according to participation in OPA into three groups. This allows to identify those adolescents who actively participate also in other forms of OPA
Results
It seems logical that adolescents with participation in three or more lessons of OPA per week, met the VPA recommendations. They already had activities scheduled and with them they could fulfill those recommendations.
Response:
In general, we agree. However, participation in OPA often might indicate a negative effect on participation in PE lessons, as OPA is often oriented more on health / recreational / relaxation, etc. tasks. Gender specifics or preferred and realized types of unorganized PA can also be significant. Therefore, analyzes of associations between participation in OPA and types of PA are important.
Discussion
The discussion seems more like an introduction to me, since there is no comparison between its results and those obtained by other researchers.
Response:
Unfortunately, such studies on this topic do not exist to compare and support or refute the obtained data. We have partially enhanced the discussion based on this and other reviewer comments.
.
Although with a different objective, in a previous publication*, they already found differences between two educational systems, which differed in physical education lessons.
Response:
The previous study took the country differences into account. However an in-depth analysis between participation in OPA and types of PA has not been performed previously. The possibility of respecting the greater focus on Polish schools on sports, greater connection with sports training (eg the system of sports schools as in the Czech Republic), the predominant single-subject PE teachers in Poland (compared to the predominant two multidisciplinary in the Czech Republic) is trying to describe the specificities of varying educational systems.
Conclussions: Participation in PELs was associated with a higher rate of meeting SPA recommendations in both countries. Compared with the Czech Republic, more PELs in the Polish education system was associated with increased daily vigorous PA and a greater portion of SPA in daily PA. Differences in overall daily and weekly moderate-to-vigorous PA between Polish and Czech adolescents were non-significant.
Response:
The results of this study correspond in some respects to previously published comparative research studies in Poland and the Czech Republic. The study addressed the association between demonstrable participation in PE lessons and daily PA, but not broader participation in OPA.
What does this new study contribute?
Response:
The study includes the role of OPA in supporting PA. The strength of this study is its investigation of the associations between OPA and the structure and intensity of weekly PA in adolescents over a period of eight years The benefits are shown in most of the monitored types of weekly PA. The contribution of the study is that it draws attention to promote increasing the weekly number of PE lessons. All this with insufficient knowledge of the level of school PE, equipment, and other consequences for increasing the number of PE lessons. It should also promote deeper extra-curricular OPA in schools, as well as effective cooperation between schools and extracurricular leisure institutions and sports clubs.
Round 2
Reviewer 2 Report
Dear authors,
After the modifications made, I think the manuscript has improved, but I still have a question after reviewing the following response to a submitted comment.
Response for authors:
“In the Polish curriculum, three PE lessons per week are compulsory for secondary schools. Exemption for health reasons or excuse from participation is possible. It is difficult to distinguish between participation in PE and other forms of OPA, so we consider it most objective to divide groups according to participation in OPA into three groups. This allows to identify those adolescents who actively participate also in other forms of OPA”
Could the exception mentioned by the authors (health reasons or excuse from participation) influence the performance of PE? That is, if the student has health problems, it may not meet the VPA recommendations. Do you think that this could lead to a significant bias in the results? Or has it been valued?
Kind regards.
Author Response
Dear authors,
After the modifications made, I think the manuscript has improved, but I still have a question after reviewing the following response to a submitted comment.
Response for authors:
“In the Polish curriculum, three PE lessons per week are compulsory for secondary schools. Exemption for health reasons or excuse from participation is possible. It is difficult to distinguish between participation in PE and other forms of OPA, so we consider it most objective to divide groups according to participation in OPA into three groups. This allows to identify those adolescents who actively participate also in other forms of OPA”
Could the exception mentioned by the authors (health reasons or excuse from participation) influence the performance of PE? That is, if the student has health problems, it may not meet the VPA recommendations. Do you think that this could lead to a significant bias in the results? Or has it been valued?
Kind regards.
Response:
Dear reviewer, this is very important point, and we would like to explain it. In informed consent participants self-reported a healthy status and no restrictions to completing the research protocol. If they declared no health problems and did not participate in PE, these are listed in group with less time participation in OPA. We valued this in results, and it did not significantly biased results. We added this statement in methods.